# Long-Haul Truck Drivers’ Perceptions of Truck Stops and Rest Areas: Focusing on Health and Wellness

**DOI:** 10.3390/ijerph21091251

**Published:** 2024-09-21

**Authors:** Fernanda Lise, Mona Shattell, Raquel Pötter Garcia, Kethelyn Costa Rodrigues, Wilson Teixeira de Ávila, Flávia Lise Garcia, Eda Schwartz

**Affiliations:** 1College of Nursing, Federal University of Pelotas, Pelotas 96077-170, RS, Brazil; kethelyn.rodrigues@ufpel.edu.br (K.C.R.); wilson.teixeira@ufpel.edu.br (W.T.d.Á.); edaschwa@gmail.com (E.S.); 2College of Health, Oregon State University, Corvallis, OR 97331, USA; 3College of Nursing, University of Central Florida, Orlando, FL 32826, USA; mona.shattell@ucf.edu; 4Human Sciences Institute, Department of Anthropology, Federal University of Pelotas, Pelotas 96010-610, RS, Brazil; raquelgarcia@unipampa.edu.br; 5College of Nursing, Federal University of Pampa, Uruguaiana 96413-170, RS, Brazil; flgarcia@ufpel.edu.br; 6College of Nursing, Federal University of Rio Grande, Rio Grande 90040-060, RS, Brazil

**Keywords:** health, behavior, work, truck driver, truck stops and rest areas, chronic non-communicable diseases

## Abstract

The work and life routine of long-haul truck drivers (LHTDs) involve the use of truck stops and rest areas to meet their basic human needs. These extensions of their workspaces on the road do not always offer adequate physical structures and services that drivers need for optimal health. This study aimed to evaluate long-haul truck drivers’ perceptions of food services, safety, physical activity, rest, and personal hygiene offered at truck stops and rest areas, as well as the correlation between these perceptions and sociodemographic, health, and work conditions variables. A cross-sectional, quantitative, and descriptive study was conducted with long-haul truck drivers from the southern region of Brazil. For data collection, a sociodemographic questionnaire and a Likert scale on food, rest, personal hygiene, safety, and physical activity services offered at truck stops and rest areas along Brazilian roads from March to August 2023 were used. The data were analyzed with simple frequency descriptive statistics. The sample consisted of 175 long-haul truck drivers. Out of these, 70.29% declared that the services of the truck stops and rest areas were charged; more than half (53.59%) of the professionals evaluated the rest service as “good” or “excellent”; the food services were “good” or “excellent” for 42.24% of the drivers. The spaces for physical activities were the worst evaluated as “bad” or “terrible” by 41.61%, followed by bathroom services (28.42%) and safety (34.24%). Rest and feeding services had better evaluations, while the services of bathroom, safety, and physical activity presented worse evaluations. Variables such as nationality, weekly working days, and marital status presented positive significance and influenced drivers’ perceptions of the services offered at truck stops and rest areas. Drivers who were Brazilian and worked more than five days a week negatively evaluated the services of rest (*p* = 0.018), safety [0.020], physical activity (0.003), and bathrooms (0.020). In addition, the physical activity services were better evaluated by single drivers than married drivers. These findings suggest that the work conditions and nationality may influence LHTDs’ perceptions of services and structures of truck stops and rest areas. These findings may reflect a lack of investments and support efforts to improve basic services such as personal hygiene, a safe environment, and physical exercises, which are fundamental to the health of the workers and aimed at reducing vulnerability and a sedentary lifestyle and meeting the basic human needs of LHTDs.

## 1. Introduction

The work of long-haul truck drivers (LHTDs) involves the transportation of products safely across the road over long distances, resulting in days, weeks, and sometimes months away from home. This work is highly stressful, lonely, and unhealthy because it involves low remuneration, absence of extensive career paths, low autonomy, and unsafe working conditions [1,2,3]. These factors negatively affect the health of these workers, contributing to the increase in chronic non-communicable diseases (CNCD) [4,5,6,7]. LHTDs face work requirements of more than 55 h a week [6] and the need to remain away from home for long periods; this forces them to use truck stops and rest areas available at gas stations located along highways to eat, sleep, rest, shower, exercise, connect to the internet to communicate with family and friends, as well as to prepare for another long day of driving [4,8,9].

Although essential, these sites do not always offer the necessary services to meet the basic human needs (BHN) of food, sleep, leisure, safety, physical comfort, personal hygiene, recreation, socialization, and physical activity for LHTDs. This situation was aggravated by the difficulty faced by LHTDs in finding parking, toilets, and food during the COVID-19 pandemic, contributing to many of them consuming more caffeine and working for longer hours [10]. Working conditions and life can affect the health of drivers [5,11]. However, few studies address the truck stops and rest areas [8,9] and explore the perceptions of workers with safe and adequate parking [8], availability of resources for personal hygiene and comfort, communication and mental stimulation, health care, safety, physical activity, and nutrition at truck stops [9].

### Challenges Related to LHTD Health and Well-Being

Upon analysis of the literature, we identified the biological, behavioral, and environmental risk factors that contribute to the development of chronic non-communicable diseases (CNCDs) in LHTDs. In addition, we present the general recommendations to promote physical, cognitive, and emotional health, which can be adapted to different cultures and health systems [4]. The main health and well-being challenges of LHTDs are related to behavior and the environment, including physical inactivity, poor diet, smoking, and chronic stress [4], which contribute to the increase in hypertension epidemics, diabetes mellitus, obesity, smoking, anxiety, depression, and post-traumatic stress disorder [1,5,6].

Previous research identified that some truck drivers’ stops in the United States did not have the necessary conditions to practice a healthy lifestyle: 50% of these locations did not offer the option of fruits, and 37% did not have fresh vegetables in the restaurant or convenience store; exercise facilities were absent in many, since 81% had no walking trail and 94% did not offer access to health care [9]. In addition, barriers to accessing health services negatively affected truckers’ health [12]. Therefore, there is an important gap in the literature related to LHTDs perceptions of food, safety, physical activity, rest, and personal hygiene services at truck stops and rest areas.

In Brazil, there are laws that protect the rights and responsibilities of LHTDs, establishing minimum standards for safety, health, and comfort in waiting and rest areas [13]. The places that can be considered as truck stops and rest areas are described in law as road stations; stops and support points; accommodation, hotels, or inns; canteens of companies or third parties; and gas stations [13]. The Special Secretariat of Social Security and Labor and the Ministry of Economy established the minimum conditions of safety, health, and comfort at truck stops and rest areas [12], and the Ministry of Infrastructure established the general procedures to recognize and issue the certification of establishments that fully meet the requirements and minimum sanitary, safety, and comfort conditions [14].

The Brazilian truck drivers surveyed about what an ideal truck stop and rest area would look like rated the following as very important: personal hygiene services (bathrooms with hot water showers and that are adapted for people with disabilities), hygiene (washing machine and dryer for laundry, utility sink for washing of other items, dumpster for trash), food (have a kitchen, stove, microwave, and space to eat food prepared in their trucks or in communal kitchens, fast food-type cafeteria, restaurant), space for rest (lounge, T.V, balcony with benches), safety (personal as well as load safety from theft and assault, surrounded parking, safety camera, entry in parking with registration or identification, lighting, maintenance of paving), communication (Wi-Fi, information about truck-stop services in applications and websites), health services (medical care), exclusive spaces for women drivers, spaces for women near convenience stores, and women’s and men’s bathrooms that are physically distant from one another [15].

Services offered at truck stops and rest areas for the satisfaction of basic human needs, prevention of CNCDs, and to promote the health and well-being of LHTDs are important. Satisfying the basic human needs and having a healthy work and rest environment contributes to achieving the goals of the Global Action Plan for the Prevention and Control of chronic non-communicable diseases 2023–2030 [16], as well as the objectives of the UN Agenda 2030 for Sustainable Development [17]. Still, given the gap in the literature about the perceptions of drivers regarding the services offered in truck stops and rest areas in Brazil, there is a need for studies that foster discussion and offer support to develop public policies that lead to the construction of adequate truck stops and rest areas that meet the health and work-life demands of these essential transportation sector workers, the LHTDs.

The results presented in this study may bring benefits to science by serving as a basis for other studies in different global regions and promote advances in driver safety, health, and well-being since the results address a topic of global interest. We present significant evidence on the influence of the work organization, especially the number of days worked, on the evaluation of truck stops and rest areas. We will support this study’s conclusions by providing evidence from previous studies and Maslow’s Theory of Human Motivation. This theory presents, at the base of a pyramid, the physiological needs essential for body homeostasis, such as water and nutrients. At higher levels are the needs of security, social esteem, and self-realization, considered fundamental for a person to feel motivated and promote his well-being and human health [18]. This study aimed to evaluate long-haul truck drivers’ perceptions about the services of food, safety, physical activity, rest, and personal hygiene offered at truck stops and rest areas in southern Brazil as well as the correlation between sociodemographic, health, and work conditions variables and these perceptions.

## 2. Materials and Methods

### 2.1. Design

This cross-sectional, quantitative, and descriptive study was conducted with LHTDs from southern Brazil, using a convenience, non-probabilistic sample with voluntary participation from drivers.

### 2.2. Ethical Considerations

This study was conducted in accordance with the Declaration of Helsinki and approved by the Ethics Committee of Federal University of Pelotas, Nursing Faculty (protocol code 5,892,602 and Certificate of Presentation for Ethics Assessment number 66722622.9.0000.5316), on 14 February 2023.

### 2.3. Sample and Setting

The inclusion criteria in this study were that the participants had to be an LHTD employed for greater than six months and could read and speak Portuguese. The exclusion criteria included not being an LHTD and not being able to read and speak Portuguese.

Data were collected in southern Brazil at several truck stops and rest areas frequented by LHTDs: a dry port characterized as the second largest in Latin America, a maritime port, three truck stops and rest points, and the waiting room of a health service of a union of road transport workers. The sample was composed of 255 drivers, of whom 175 were long-haul truck drivers; the others were excluded as they transported people or cargo in the urban and interurban sectors, not making use of the truck stops and rest areas used by LHTDs.

### 2.4. Data Collection

#### Procedure

The participants were approached in person by one of the researchers, accompanied by four nursing master’s students and five academic researchers (four from the nursing discipline and one from psychology), all of whom had previously received guidance and ongoing training on data collection procedures. Upon contact with each potential participant, researchers explained the objectives of the study and the type of participation desired. the Free and Informed Consent Form was read, and two forms were signed; one was given to the participant, and the researcher kept the other. Data collection took place from March to August 2023.

### 2.5. Instruments

A sociodemographic form with 25 questions and an instrument with 6 items was used to evaluate the perceptions of LHTDs on the services of rest, food, toilets, safety, and physical activity offered at truck stops and rest areas in Brazil as well as if the service was free or paid. This instrument was developed by the authors, supported by an extensive literature review [4], after consultation with health specialists, LHTDs, and members of the Brazilian road transport workers unions. It uses a Likert-type scale with 6 items about the rest, food, toilets, safety, and physical activity service as well as about the payment of the service (free or paid). The participants were asked to rate the services as terrible, bad, average, good, or excellent, including an option of service unavailable in truck stops and rest areas frequented by LHTDs. Instruments were presented to each LHTD to complete in paper and pencil; however, some participants needed support in reading the items, likely due to literacy levels.

A field notebook was used to record the researchers’ perceptions about the environment, the organization of health care services, ports, and stopping and resting points where data collection took place, and it was used to support the description of the data and the data analysis.

### 2.6. Data Analysis

The data were entered in a Google form and then analyzed from the Excel for Windows^®^ spreadsheet file. For each variable analyzed, the value observed in the sample (n) and the percentage (%) were expressed. The data were analyzed using the Statistical Analysis Software (SAS, Version 9.4) from a database built through Excel. To measure the differences among the scores attributed to the different services offered by truck stops and rest areas, the sum score was used for each variable, and the Wilcoxon–Mann–Whitney or Kruskal–Wallis (non-parametric Anova) test was applied. This was followed by the post hoc test of multiple comparisons of Dunn in cases where the observed variable had more than two categories. A confidence level of 95% was considered (α = 0.05).

## 3. Results

The study sample included 175 LHTDs, all of whom were men (100%) who had an average age of 43 years; they were mostly Brazilian (86.2%), and some were Argentinian (10.3%) and Chilean (3.4%). The percentage of LHTDs who worked more days a week was higher for Chileans and Argentineans than Brazilians. On the other hand, the percentage of LHTDs driving more than 9 h per day was higher for Chileans, followed by Brazilians. While 83% of Chileans and 61% of Argentineans work more than 6 days a week, 58% of Brazilians work fewer days per week (up to 5 days). As for the hours of driving, 66% of Chileans, 59% of Brazilians, and 50% of Argentineans drove between 9 and 12 h a day. Among the participants, 100% of Chileans and Argentineans made international routes, and 58% of Brazilians made, in addition to national, international routes through Latin America (Table 1).

The analysis of the sample of LHTDs found that approximately one-third (29.71%) stopped and rested at areas where services were offered free of charge; the remaining 70.29% stopped at areas that charged for their services. As for the quality of services that were offered at either paid or free locations, rest was perceived by more than half (50.8%) as “good” or “excellent”, followed by “average” by 30.2%, and the food services were perceived as “good” or “excellent” by 38.8%, followed by “average” by 46.2%; the other services were perceived with less satisfaction. The bathroom services were considered “average” by 37.7%, followed by “good” or “excellent” by 32.0%. Security services were perceived as “average” by 35.4%, followed by “terrible” or “bad” by 35.4%. The spaces for physical activities were the worst evaluated as “bad” or “terrible” by 31.4%, followed by “unavailable” by 26.0% (Figure 1).

Significant differences were found in the evaluation of rest, safety, toilets, and physical activity services. Marital status, working days in the week, and nationality were the variables that most influenced the perceptions of drivers. The spaces for rest were best evaluated by the drivers who worked up to five days a week (*p* = 0.0180) and those who were Chilean and Argentinean (*p* = 0.0003). Security services was best evaluated by drivers who were single (*p* = 0.0068), those who worked up to five days a week (*p* = 0.0200), and those who were Chilean and Argentinean (*p* = 0.0001). Drivers working up to five days a week (*p* = 0.0200) as well as Chilean and Argentinean drivers evaluated personal hygiene services (bathrooms) more positively (*p* = 0.0003). The safety of stopping and resting places was rated highest by single drivers (*p* = 0.0068) as well as Chilean and Argentinean drivers (*p* = 0.0001). Single drivers (*p* = 0.0016), those who worked up to five days a week (*p* = 0.0036), as well as Chilean and Argentinean drivers evaluated the spaces for physical activities more positively (*p* = 0.0004). No significant differences were found in the evaluation of the quality of food services. Age, health problems, and the number of working hours per day did not influence the evaluation of the services (Table 2).

## 4. Discussion

This study presents the perceptions of the Brazilian, Chilean, and Argentinean LHTDs about the services offered at truck stops and rest areas in Brazil. This is the first study to examine LHTDs’ perceptions of truck stops and rest areas in Brazil. Truck stops and rest areas are viewed as part of the occupational work environment of these workers. This study found that the satisfaction with physical activity, security, and bathroom services was worse than the satisfaction with rest and food services.

The spaces for the practice of physical activities were perceived as “bad” or “terrible” by 30.4% of drivers, followed by “unavailable” by 27.7%. The analyses of marital status, amount of work days per week, and nationality showed that Brazilian LHTDs, married or separated, who work more than five days each week perceived these services as worse than the other LHTDs. This negative evaluation may be related to the quality of the structures at truck stops and rest areas for the practice of physical activity. Single LHTDs, possibly younger people, feel a lesser need for spaces suitable for physical activity because they have greater physical resistance and fewer effects of occupational and/or chronic diseases. For drivers who work up to five days a week, physical activity is possibly being neglected due to a lack of incentive, which leads to sedentary behavior, and/or due to a high workload that acts as a barrier to physical activity. As for nationality, one can consider the influence of cultural factors and the quality of the physical activity services in their home countries, which means the services on Brazilian highways were better evaluated by Chilean and Argentinean drivers. The lack of adequate space for physical activities may contribute to sedentary behavior and, consequently, to the development of chronic diseases such as obesity, hypertension, and diabetes. Previous studies have shown that drivers practice little physical activity [19,20], and 90% of them had a low muscle mass index [19], possibly due to a high workload and fewer opportunities to exercise.

Physical exercise is a physiological need, laid out at the base of the hierarchy of needs pyramid [18] and has the following benefits: the reduction of stress and the risk of chronic diseases; improved sleep quality; increased strength and flexibility; and physical and emotional balance [21]. The difficulty in satisfying the need for physical activity can trigger feelings of frustration and an increase in depressive symptoms, obesity, hypertension, and/or diabetes. To make truck stops and rest area environments healthier, several researchers recommend the installation of gyms, availability of hiking trails, and physical activity with supervision of health professionals at stopping and resting points [5,21].

Security services were perceived as “average” by 35.1%, followed by “bad” or “terrible” by 33.5% of LHTDs. The analyses of the work days per week and the nationality showed that Brazilian LHTDs who work more than five days each week presented the worst perception of this service. This result may be related to sociocultural factors, the organization of the work process and services, urban violence, and the restriction at the opening hours of services at truck stops and rest areas. Safety may have different meanings; however, when referring to truck stops and rest areas, it is understood that safety should be related to aspects involving the physical structure of truck stops and rest areas, such as an opening period of 24 h for 7 days a week, lighting, paving, signaling, access ramps for wheelchair users, fire extinguishers, internet access, parking area and exclusive supply for the truck, enclosed space, monitoring by cameras, and surveillance service aiming to safeguard users, trucks, and cargo. Urban violence is a social phenomenon, and this may have influenced the negative perception by drivers, considering that truck stops and rest areas do not always present the minimum conditions described in the legislation, which includes offering electronic surveillance or monitoring, as well as the availability of enclosed parking with access control, with increased safety in those that require payment for the parking service [12]. A previous study revealed the difficulties faced by North American drivers to find parking spaces on weekdays as well as the importance of internet access, considered very important by 51% of drivers [8].

Safety is a construct that is at the base of the hierarchy of needs pyramid, arranged in the second level, after physiological needs. It involves individual security as well as the security of body, employment, resources, housing, family, and health against dangers and threats [18]. When this need is not met, it leads to a state of stress due to the feelings of insecurity and anxiety. This further triggers neurophysiological changes, which lead to increased heart beats, blood pressure, and blood glucose levels, as well as sleep disturbances and difficulty in concentration. A previous study showed that post-traumatic stress disorder was a predictor of the use of mental health care among LHTDs [22]. Thus, the working and rest environment needs to provide safety to preserve the physical and mental health of LHTDs.

The perception of drivers regarding bathroom services was “average” by 36.6%, followed by “good” or “excellent” by 35.1%. The analyses of the working days per week and the nationality revealed that Brazilian LHTDs, who worked more than five days each week, presented the worst perception of this service. This result is possibly related to the organization of the truck stops and rest areas’ services, with the restriction of opening hours of personal hygiene services and/or noncompliance with the rules established by the Brazilian government. In addition, cultural factors may have influenced the perceptions of the quality of services by foreign (Chilean and Argentinean) drivers. This service deals with the sanitary facilities and a personal hygiene space, with basic objects and items such as sinks, taps, soap, mirrors, toilets, showers with hot and cold water, toilet paper, female absorbents, paper towels or hand dryers, trash bins, lighting, and space to store belongings while using these services. In Brazil, the minimum conditions for the services of truck stops and rest areas, presented in Ordinance 1.343/2019, also provide that the bathrooms are separated by sex and that they are individualized and sanitized [12]. This personal hygiene space, in addition to containing the basic items, is a space for satisfying a basic human need [18] and needs to be easily accessible. The absence of hot water showers in the southern region of Brazil, where winter is usually severe, with very low temperatures, was mentioned by some drivers when completing the form. Failure to meet with the minimum conditions affects physiological elimination, self-care, security, and privacy, and therefore the structure needs to offer an environment that meets these needs and is clean and pleasant to ensure safety and preserve the health of users. A study conducted in the United States revealed that drivers considered this service very important (62%) [8].

The act of resting encompasses another basic need, sleep. Resting and sleeping are considered physiological needs, arranged at the base of the hierarchy of needs pyramid because they are fundamental for biological, social, and physical balance [18]. The rest service was perceived as “good” or “excellent” by more than 50% of drivers, followed by “average” by another 29.8%. The analyses of the working days per week and the nationality showed that Brazilian LHTDs who work more than five days each week presented the worst perception of this service. Cultural and work environment organization factors may have influenced this result, making foreigners evaluate the structure of the rest service more positively; those that work for more than five days have a negative perception of the quality of service. This result may be related to the understanding that most riders consider the truck as a rest space where they usually sleep [23].

The perception about this service involves other aspects, possibly disregarded by the drivers, that go beyond a calm and quiet environment for rest in the truck, which is fundamental for good rest during breaks, and these rest breaks from driving are essential. Rest breaks have been found to be associated with a reduction in near-miss accidents [24]. Quality and quantity of sleep are also important for drivers and roadway safety. One study found that poor sleep hygiene was considered a predictor of crashes [25] and stress at work [5]. The rest service, which can provide a place for rest or sleep, can also include the environment outside the workspace, in this case, outside the truck, which allows contact with nature and socialization, such as the availability of spaces with a green area, benches, networks, games, TV, internet access, and hotel service. A previous study revealed that internet access was considered very important by 51% of drivers [8]. Promoting adequate rest spaces has the potential to decrease the use of strategies like caffeine to mitigate sleepiness at the wheel [26]. Promoting rest spaces may also prevent the risk of cardiometabolic diseases, obesity, and cancer as well as reduce social isolation and stress, thus increasing the well-being [10,21,27] and mental health of LHTDs [5,22,28]. This would also reduce fatigue-related crash risk [29,30] for LHTDs and for others who share the road.

The perception regarding the food service was perceived as “good” or “excellent” by 44.0%, followed by “average” by 41.4% of the drivers. This result may be related to the practice of drivers preparing their own meals in the kitchen boxes of their truck, as mentioned by some participants, which were considered healthier. On the other hand, some drivers eat ready-made snacks or dinner at truck stop and rest areas’ restaurants. Previous studies have shown that 73% of drivers used to eat food brought from home, while 37.0% regularly or always ate food served at truck stops. To conserve and prepare food, 94% had a refrigerator, 62% had a gas pot, and 8% had a microwave in the truck. The items brought from home were fruits (62%), sausages (50.6%), sandwiches (38.7%), pre-ready meals (37%), sweets (35.4%), and raw vegetables (31%) [31]. As for the food quality, 88% consumed fewer vegetables than recommended, 63% consumed at least one health-damaging food per day, and 65% consumed a can of sugary drink per day [19]. It is understood that despite the positive perception of food in Brazilian truck stops and rest areas, drivers reported informally, when completing the form, that the food options available in these places often had few or no fruits and salads, offering only options of ultra-processed foods, rich in saturated fat, or fast food. The literature showed a significant correlation between drivers who eat meals in restaurants and increased weight [10]. Thus, it is understood that malnutrition is related to the quality of the food available for consumption in truck stops and rest areas and increases the risk of becoming overweight or obese and developing other cardiometabolic diseases related to diet, bringing irreparable consequences to people, families, and communities [32].

The choice of high-energy foods may be related to the time spent on work and the lack of it to prepare and/or hold balanced meals. Studies have shown that the daily workload of LHTDs ranged from more than 10 h (53%) [22] to more than 13 h per day (37.5%) [6]. Thus, the organization of work can compromise the adoption of healthy behaviors by not respecting the human right to adequate food and satisfying the basic human need for quality food. Feeding is a physiological need, arranged at the base of the hierarchy of needs pyramid [18]. As a result, poor diet can affect the driver’s health by contributing to increased cholesterol levels, blood pressure, and diabetes [22]. Another factor that can influence the perception of food service is the quality of the environment to have the meal, which needs to be clean, comfortable, and furnished with tables and chairs, as recommended by the Ministry of Economy and Labor Brazil in Ordinance 1.343/2019 [12]; thus, adopting healthy behavior requires structural changes that provide quality meals to LHTDs.

The payment requirement to use parking and personal hygiene services was reported by 68.1% of LHTDs. Brazilian legislation provides that the services of personal hygiene and parking are offered free of charge to users, and if a fee is required for the parking of the vehicle, it must be surrounded and have access control (Ordinance 1./2019) [12]. However, this is not the reality of most truck stops and rest areas. In this sense, charging for these services can be stress triggering for LHTDs since it can compromise the security and finances of the worker. The physiological needs of physical protection and search for shelter are at the base of the hierarchy of needs pyramid [18], and in case of impossibility of payment, the driver will need to seek another place to park after long hours of work. In addition, when facing difficulties in accessing other services essential for the satisfaction of basic human needs such as personal hygiene, elimination, comfort, and rest, drivers can present with an impairment of their health and well-being, such as that occurred during the COVID-19 pandemic [10].

Given the cultural differences between the regions of Brazil and South America, the results cannot be generalized to other regions; however, they should serve as a basis for other studies in different countries to evaluate the drivers’ perceptions of work environment services for basic necessities such as food, bathrooms, and space for physical activity. One study found that no truck stops offered exercise facilities, 94% lacked access to health care, 81% lacked a walking path, 50% lacked fresh fruit, and 37% lacked fresh vegetables in their restaurants or convenience stores, in truck stops and rest areas located in the US [9].

The results of our study reported here may present constructs of global interest related to food, bathrooms, rest, safety, and physical activity in truck stops and rest areas located in Brazil and the United States (US) to better address the health needs of this underserved population of workers.

### 4.1. Limitations

There are some limitations that need to be taken into account when interpreting the present findings. First, in view of the absence of validated instruments to evaluate the perceptions and satisfaction of LHTDs with the services offered at truck stops and rest areas, in this study, an instrument developed by the authors was used and not validated to collect the data. Second, given the difficulty in accessing participants through transport companies, the sample selection was, for convenience, performed by face-to-face approaches in truck stops and rest areas. Third, the results of the perceptions of LHTDs were compared with sociodemographic data, health problems, number of days and hours of work per day, and nationality of the driver, but did not take into account other factors such as rest breaks, statutory rest breaks, and variables related to sleep and drowsiness, such as the use of stimulating substances (sleep hours, obstructive sleep apnea, and insomnia). These would be important areas for future research. Despite these limitations, we emphasize that they do not reduce the quality of the study and implications of the results. LHTDs need adequate services on the road, if not for these workers than for the enhanced road safety that is associated with LHTDs who take regular rest breaks and have good sleep hygiene. Workplace and government policies should provide greater support to improve the quality of these settings.

Validated instruments to study these work settings is an area for future research. In addition, samples obtained through stratified sampling that also studies related issues relevant to driving safety may increase the rigor, reliability, and generalizability of the results. Although this study is the first to evaluate the perceptions of LHTDs regarding the services offered at truck stops and rest areas in southern Brazil, and the results indicated aspects of these areas with the greatest needs, other studies from other regions of Brazil and South America are necessary to better understand this phenomenon among road-transport workers.

### 4.2. Contributions to Practice

It is expected that the results presented here may contribute to supporting the health care of drivers and preventing chronic non-communicable diseases, as well as serve as the basis for other studies and the development of public policies to advance the quality of services offered at truck stops and rest areas in Brazil, aiming at the security, well-being, and health of LHTDs.

## 5. Conclusions

The perceptions of LHTDs about the services of food, safety, physical activity, rest, and bathroom offered at truck stops and rest areas in Brazil showed that the services of rest and food were considered good or excellent; however, the services of bathroom, safety, and physical activity were not well evaluated. It was evident that marital status, the number of working days per week, and nationality were the variables that influenced the drivers’ perceptions of rest, safety, physical activity, and bathroom services. As it comes to the assessment of perception, it is not possible to generalize this result; however, it represents the experience of drivers and indicates the need for investments to improve the provisions of basic services, such as personal hygiene, safe environment, and physical exercises meeting basic human needs, aiming to reduce vulnerability and sedentary lifestyles, fundamental to the prevention and control of chronic non-communicable diseases, and to promote the well-being and health of LHTDs.

## Figures and Tables

**Figure 1 ijerph-21-01251-f001:**
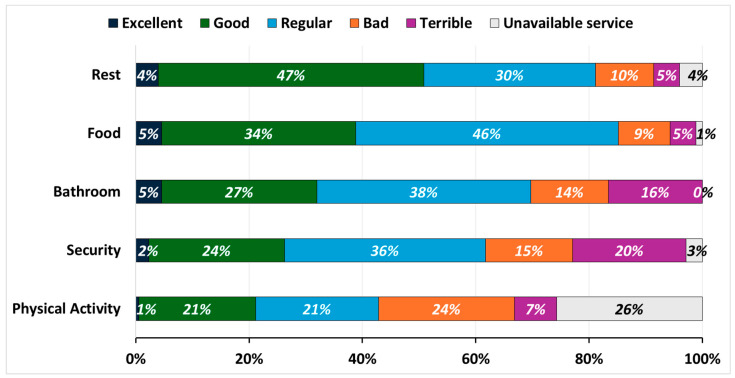
Truck drivers’ perceptions of services at truck stops and rest areas, Brazil, 2023.

**Table 1 ijerph-21-01251-t001:** Sociodemographic characteristics of long-haul truck drivers, Brazil, 2023.

Variables	n	%
**Sex**		
Male	**175**	**100**
Female	00	0.00
**Age**		
Up to 39 years	68	39.08
40 years or +	107	60.92
** *Average: 43.75; DP: 11.86* **		
** *Marital status* **		
Married/Stable Union	121	69.14
Separated/widowed	43	24.57
Single	11	6.28
** *Education* **		
Fundamental incomplete	32	18.28
Fundamental complete	40	22.85
High school	99	56.57
Higher education	4	2.28
**Nationality**		
Brazil	151	86.21
Argentina	18	10.34
Chile	6	3.45
**Routes**		
National and international	126	71.46
National	49	28.00
**Hours worked per day**		
Up to 8 h	35	20.00
9 to 12 h	100	57.14
More than 12 h	40	22.85
**Working days per week**		
Up to 5 days	95	54.28
6 or more days	80	45.72
**Free truck stop service**		
Yes	52	29.71
No	123	70.29

**Table 2 ijerph-21-01251-t002:** Correlations between the sociodemographic, health, and work conditions variables and the perceptions of LHTDs on the services of rest, safety, physical activities, bathrooms, and food at truck stops and rest areas in Brazil, 2023.

Variables	Services									
	Rest		Security		Physical Activity		Bathroom		Power Supply	
	MS	*p*	MS	*p*	MS	*p*	MS	*p*	MS	*p*
Age										
>40 years	95.3	0.050	88.0	0.497	90.8	0.258	84.7	0.238	90.8	0.258
40 years or more	83.3		87.9		86.1		90.0		86.1	
**Civil status**										
Married/ Stable union	84.2		85.0		82.7		87.2		90.1	
Separate	89.2	0.260 ^a^	57.8	0.006 ^a^	62.3	0.00 *^a^	81.8	0.774 ^a^	64.3	0.191 ^a^
Single	98.0		104.6		109.7		91.9		88.5	
**Health problems**										
No	89.5	0.233	88.9	0.329	88.9	0.331	90.4	0.135	88.7	0.370
Yes	83.6		85.2		85.2		81.2		86.0	
**Working days in a week**										
Up to 5 days	99.6	0.018 *	94.5	0.020 *	103.8	0.003 *	92.8	0.020 *	89.2	0.413
5 days<	83.2		85.2		81.4		86.0		87.5	
**Hours worked per day**										
Up to 8 h	80.8	0.216	88.9	0.461	76.5	0.116	93.3	0.287	85.9	0.410
<8 h	89.0		87.8		89.7		87.2		88.3	
**Nationality**										
Brazilian	83.0	0.000 *	82.3	0.000 *	82.8	0.000*	82.8	0.000 *	85.9	0.080
Foreign	117.8		121.9		118.8		118.8		100.2	

Note: MS: Mean score; *p*-value * *p* < 0.05 was considered significant. ^a^: Kruskal-Wallis test was performed to analyze the statistical significance.

## Data Availability

The data presented in this study are available on request from the corresponding author upon reasonable request.

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
