# Peer review of "Long-Haul Truck Drivers’ Perceptions of Truck Stops and Rest Areas: Focusing on Health and Wellness"

_ijerph, 2024, doi:10.3390/ijerph21091251_

Round 1

Reviewer 1 Report

Comments and Suggestions for Authors

Thanks for submitting this paper for being considered in International Journal of Environmental Research and Public Health. The manuscript (ijerph-3146788) is empirical study study examining the perceptions of long-haul truck drivers about the services offered at truck stops and rest areas in Brazil. The topic addressed is worth of investigation, and the theoretical/empirical approaches considered result interesting and may contribute in a substantial manner to the importance of services offered at truck stops and rest areas for the satisfaction of basic human needs, prevention of chronic diseases and to promote the health and wellbeing of LHTDs.

 My comments are listed below:

1. The authors should conduct a more detailed literature review. Besides defining the problem in the introductory part, it is necessary to analyze the existing scientific material on the topic of attitudes and other personal attributes of truck drivers on human needs, chronic diseases, safety challenges, and the like. The authors have not defined the research gap or the scientific contribution. The manuscript is reduced to descriptive statistics, which in my opinion has very little scientific contribution.

2. The topic addressed by the authors is of exceptional importance for the safety of truck drivers in traffic, as well as the overall health profile of this population. However, the methodological approach used in the paper has two main shortcomings, which significantly affect the validity of the research and the ability to generalize the obtained results. First of all, the results are based on a convenience, non-probabilistic sample, whose representativeness is questionable, and the derived conclusions are very limited. Secondly, the LHTD measuring instrument that the authors developed is highly questionable. Did the authors conduct a process to determine the validity and reliability of the LHTD measuring instrument? If they did, it is important to present its psychometric properties. This cannot be determined from the manuscript.

3. Please explain the meaning of the abbreviation NCCD, at the first place it is mentioned in the paper (i.e. page 2, line 90).

Despite the very limited results of the given research, I suggest the authors do not give up on this topic, as it is of exceptional importance for the safety and health condition of heavy truck drivers.

Author Response

Reviewer 1

  1. The authors should conduct a more detailed literature review. Besides defining the problem in the introductory part, it is necessary to analyze the existing scientific material on the topic of attitudes and other personal attributes of truck drivers on human needs, chronic diseases, safety challenges, and the like. The authors have not defined the research gap or the scientific contribution. The manuscript is reduced to descriptive statistics, which in my opinion has very little scientific contribution.

Authors: Thank you for the feedback. We added a review of the literature (lines 80 – 96), the problem (lines 61-79), scientific contribution (lines 133-137), and the gap in research (lines 94-96). In this revised manuscript, we present the analysis of the correlation between sociodemographic, health, and work conditions variables and the LHTDs perceptions about the services of food, safety, physical activity, rest, and personal hygiene offered at truck stops and rest areas in Southern Brazil (lines 144-147).

  1. The topic addressed by the authors is of exceptional importance for the safety of truck drivers in traffic, as well as the overall health profile of this population. However, the methodological approach used in the paper has two main shortcomings, which significantly affect the validity of the research and the ability to generalize the obtained results. First of all, the results are based on a convenience, non-probabilistic sample, whose representativeness is questionable, and the derived conclusions are very limited. Secondly, the LHTD measuring instrument that the authors developed is highly questionable. Did the authors conduct a process to determine the validity and reliability of the LHTD measuring instrument? If they did, it is important to present its psychometric properties. This cannot be determined from the manuscript.

Authors: Thank you for the feedback. We conducted new data analysis and presented the correlation between sociodemographic, health, and work conditions variables and the LHTDs perceptions about the services of food, safety, physical activity, rest, and personal hygiene offered at truck stops and rest areas (lines 220 -227). The instrument was developed and supported through an extensive literature review, and in consultation with health specialists, LHTDs, and members of the Brazilian road transport workers unions (lines 207-209). No validated measurement tools were found.

  1. Please explain the meaning of the abbreviation NCCD, at the first place it is mentioned in the paper (i.e. page 2, line 90).

Authors: Chronic Non-Communicable Diseases (CNCD) (line 22).

Reviewer 2 Report

Comments and Suggestions for Authors

Abstract: the objective must be explicit and in line with the objective presented at the end of the introduction. Present the type of study with specifications, not just that it is quantitative. The questionnaire used is validated? The result does not present the end of the statistics, review this analysis. The conclusion should respond to the objective.

Introduction

3rd paragraph too long

Remove the hypothesis from the last paragraph

Present the gap in the literature and how the study can be applied and generate benefits, link between science and community.

Methods

The study can spread the results to other parts of the country and other countries? Each region has a tradition of rest stops for drivers? Different eating patterns? Different standards of physiological needs services (bathrooms)

Age and length of service can be biased, how did they work on this? Sex too

The questionnaire was tested? It underwent refinement? Analysis by experts? It did a pre-test? The statistical analysis is very fragile and does not allow for in-depth analysis, with comparisons, correlations, inferences, association. Perform robust statistical analysis.

Results:

do again after statistical analysis

Discussion:

do again after statistical analysis

Conclusion:

do again after statistical analysis

Comments on the Quality of English Language

It ok in my opinio, easy to read.

Author Response

Reviewer 2

Abstract: the objective must be explicit and in line with the objective presented at the end of the introduction. Present the type of study with specifications, not just that it is quantitative. The questionnaire used is validated? The result does not present the end of the statistics, review this analysis. The conclusion should respond to the objective.

 Authors: Thank you for the feedback. We changed the objectives in the abstract and in the end of the introduction. We conducted a new analysis of the correlation between sociodemographic, health, and work conditions variables and the LHTDs perceptions about the services of food, safety, physical activity, rest, and personal hygiene offered at truck stops and rest areas in Southern Brazil (lines 144-147). We added the type of study as suggested (line 32). The instrument was developed and supported through an extensive literature review, and in consultation with health specialists, LHTDs, and members of the Brazilian road transport workers unions (lines 207-209). No validated measurement tools were found. We presented new results based on our new analysis. Brazilian drivers and those who worked more than five days a week negatively evaluated the services of rest [p = 0.018], safety [0.020], physical activity [0.003] and bathrooms [0.020]. In addition, the physical activity service was better evaluated by single drivers than those who were married (lines 43-49 and 258-275).

Reviewer 2

Introduction

3rd paragraph too long

Remove the hypothesis from the last paragraph

Present the gap in the literature and how the study can be applied and generate benefits, link between science and community.

  Authors: Thank you for the feedback. We shortened the third paragraph and removed the hypothesis.  We added the scientific contribution (lines 133-137) and gap in the literature (lines 94-96).

Reviewer 2

Methods

 The study can spread the results to other parts of the country and other countries? Each region has a tradition of rest stops for drivers? Different eating patterns? Different standards of physiological needs services (bathrooms)

Age and length of service can be biased, how did they work on this? Sex too

The questionnaire was tested? It underwent refinement? Analysis by experts? It did a pre-test? The statistical analysis is very fragile and does not allow for in-depth analysis, with comparisons, correlations, inferences, association. Perform robust statistical analysis.

 Authors: Thank you for the feedback. We conducted new data analysis and presented the correlation between sociodemographic, health, and work conditions variables and the LHTDs perceptions about the services of food, safety, physical activity, rest, and personal hygiene offered at truck stops and rest areas (lines 220-227). We added a discussion of cultural differences (lines 429-437). The instrument was developed and supported through an extensive literature review, and in consultation with health specialists, LHTDs, and members of the Brazilian road transport workers unions (lines 207-209). No validated measurement tools were found.

Reviewer 2

Results:

do again after statistical analysis

 Authors: Thank you for the feedback. We conducted a new data analysis and presented the correlation between sociodemographic, health, and work conditions variables and the LHTDs perceptions about the services of food, safety, physical activity, rest, and personal hygiene offered at truck stops and rest areas (lines 258-272).

Reviwer 2

Discussion:

do again after statistical analysis

 Authors: Thank you for the feedback. We added text to the discussion section about the new results from the correlations between sociodemographic, health, and work conditions variables and the LHTDs perceptions about the services of rest (lines 288-290), safety (lines 292-299), physical activity (lines 313-347), and personal hygiene (lines 366-371).

Reviewer 2

Conclusion: do again after statistical analysis

 Authors: Thank you for the feedback. We revised the conclusion section based on the new results from the new analysis (lines 458 – 460).

Round 2

Reviewer 1 Report

Comments and Suggestions for Authors

Dear Authors, Thank you for the submitted review and the extensive revisions you made to the manuscript. Although I still believe that the manuscript has serious methodological limitations regarding the formation of the questionnaire and the sampling method, due to the significance of the topic and the pioneering endeavor in this field, as well as the efforts of the authors, I believe the manuscript should be accepted. I wish you all the best in your future work.

Author Response

Dear reviewer, we thank you for the feedback.
